# High-Frequency Ultrasound Imaging to Distinguish High-Risk and Low-Risk Dermatofibromas

**DOI:** 10.3390/diagnostics13213305

**Published:** 2023-10-25

**Authors:** Danhua Li, Feiyue Yang, Yang Zhao, Qiao Wang, Weiwei Ren, Liping Sun, Dandan Shan, Chuan Qin

**Affiliations:** 1Department of Medical Ultrasound, Jinshan Hospital, Fudan University, Shanghai 201508, China; denise2786@126.com (D.L.); 18121152297@163.com (Y.Z.); 2Department of Medical Ultrasound, Shanghai Tenth People’s Hospital, Tongji University, Shanghai 200072, China; yangfeiyue1128@163.com (F.Y.); wangqiao078@163.com (Q.W.); rww9456@163.com (W.R.); sunliping_s@126.com (L.S.); 3Shanghai Engineering Research Center of Ultrasound Diagnosis and Treatment, Shanghai 200072, China; 4Department of Medical Ultrasound, Shanghai Skin Disease Hospital, Ultrasound Research and Education Institute, Tongji University School of Medicine, Shanghai 200443, China; 5Department of Ultrasound, Karamay Central Hospital, Karamay 834000, China

**Keywords:** dermatology, fibrous histiocytoma, ultrasonography, diagnosis

## Abstract

Background: Dermatofibroma has various pathological classifications, some of which pose a risk of recurrence and metastasis. Distinguishing these high-risk dermatofibromas based on appearance alone can be challenging. Therefore, high-frequency ultrasound may provide additional internal information on these lesions, helping to identify high-risk and low-risk dermatofibroma early. Methods: In this retrospective study, 50 lesions were analyzed to explore the correlation between clinical and high-frequency ultrasound features and dermatofibroma risk level. Based on their pathological features, the lesions were divided into high-risk (*n* = 17) and low-risk (*n* = 33) groups. Subsequently, an identification model based on significant high-frequency ultrasound features was developed. Results: Significant differences were observed in the thickness, shape, internal echogenicity, stratum basal, and Doppler vascular patterns between the high-risk and low-risk groups. The median lesion thickness for the high-risk dermatofibroma group was 4.1 mm (IQR: 3.2–6.1 mm), while it was 3.1 mm (IQR: 1.3–4.2 mm) for the low-risk dermatofibroma group. In the high-risk dermatofibroma group, irregular morphology was predominant (70.6%, 12/17), the most common being dermis-to-subcutaneous soft tissue penetration (64.7%, 11/17), and heterogenous internal echogenicity was observed in the majority of cases (76.5%, 13/17). On the other hand, regular morphology was more prevalent in the low-risk dermatofibroma group (78.8%, 26/33), primarily limited to the dermis layer (78.8%, 26/33), with homogeneous internal echogenicity also being prevalent in the majority of cases (81.8%, 27/33). Regarding the Doppler vascularity pattern, 69.7% (23/33) of low-risk dermatofibromas had no blood flow, while 64.7% (11/17) of high-risk dermatofibromas had blood flow. Conclusion: High-frequency ultrasound is crucial in distinguishing high-risk and low-risk dermatofibromas, making it invaluable for clinical management.

## 1. Introduction

Dermatofibroma (DF), or fibrous histiocytoma, is a typical soft tissue lesion of the skin, accounting for approximately 3% of dermatologically excised specimens [1]. Typical dermatofibroma (DF) accounts for roughly 45% [2], which is easy to diagnose [3,4]. Most DFs rarely reoccur even if incompletely removed; thus, they are defined as low-risk dermatofibromas (DFs) [5]. However, it has been confirmed that certain classifications exhibit a higher propensity for metastasis (deep DFs) and local recurrence (aneurysmal DFs, atypical DFs, and cellular DFs), with recurrence rates greater than 20% [6,7]. These are defined as high-risk DFs.

Due to their metastatic and recurrent nature, high-risk DF patients require a prompt diagnosis and extensive resection. Conversely, low-risk DF patients may be treated with more conservative modalities, such as liquid nitrogen cryotherapy or laser therapy. There are differences in the management measures between high-risk and low-risk DF. Therefore, it is crucial to stratify the DF risk. However, an accurate diagnosis is challenging due to the diverse clinical presentations of these DFs.

It is well known that a pathological examination is the gold standard for the classification of DFs. However, due to the limited evaluation range, the aggressive part of the tumor can be missed, without which comprehensive spatial information about the lesion’s size, thickness, and infiltration depth cannot be gained. Moreover, biopsy is an invasive procedure [8,9].

With advances in technology, noninvasive examinations such as dermoscopy, optical coherence tomography (OCT), reflectance confocal microscopy (RCM), and high-frequency ultrasound (HFUS) are being increasingly used in relation to various skin diseases [10]. However, dermoscopy only allows the superficial structure of the lesion to be observed, while information about the deep part of the lesion cannot be explored. Although optical coherence tomography (OCT) and reflectance confocal microscopy (RCM) can provide high-resolution morphological information about the lesion, their penetration rate is relatively low, and they are limited to visualizing only the epidermis and superficial dermis [9,11]. Therefore, these techniques make distinguishing many DFs with basal layers in the subcutaneous tissue layer difficult.

On the contrary, high-frequency ultrasound (HFUS) reasonably balances penetration and resolution while displaying lesion depth information. Furthermore, it is advantageous because of its inexpensive, noninvasive, reproducible, and real-time imaging. The utilization of high-frequency ultrasound enables the stratification of superficial tissues, thereby facilitating the standardization of “ultrasound anatomy”. This standardized approach aids in determining whether dermatofibroma necessitates an upgrade and consequently assists clinicians in formulating a precise treatment plan [12]. Therefore, we hypothesized that HFUS might be a reliable tool for diagnosing suspected high-risk DF and low-risk DF and differentiating them when noninvasive [10,13,14].

To the best of our knowledge, the value of HFUS in differentiating high-risk DF from low-risk DF has not been evaluated sufficiently. Therefore, we conducted this study to explore this and developed a practical discrimination model for differential diagnosis.

## 2. Materials and Method

This study was approved by the hospital ethics committee, and informed consent was waived due to the retrospective nature of the study. In addition, all procedures were conducted in accordance with the Declaration of Helsinki.

### 2.1. Patients

From April 2017 to October 2022, a retrospective review was conducted on patients who underwent an ultrasound examination and surgical resection at Jinshan Hospital of Fudan University and Shanghai Skin Diseases Hospital. The images obtained were stored in a centralized database. By reviewing the pathological slides of the enrolled cases, a second reading was performed to classify the DF subtypes further. Among them, aneurysmal, cellular, and deep DFs were defined as high-risk DFs, while other types, such as classic, atrophic, epithelial, and atrophic DFs, were defined as low-risk DFs.

The inclusion criteria were as follows: (a) the diagnosis of DF was confirmed by pathological examination, and (b) the lesion received a high-frequency ultrasound (HFUS) scanning. The exclusion criteria were as follows: (a) the quality of ultrasound images was poor, and (b) the DF lesion received treatment prior to the HFUS examination. Finally, 49 patients with 50 DFs were enrolled in this study. Among them, one patient had two lesions (Figure 1).

The clinical features of the patients were collected, including age, sex, lesion location, and the presence of multiple DF (defined as more than two DF lesions).

### 2.2. HFUS Scanning

All ultrasound scanning was performed by the same sonographer (Q. Wang) with more than eight years of experience in dermatologic ultrasounds. The equipment used for the HFUS examination of skin lesions was as follows: (a) an HFUS linear-array transducer (4–20 MHz, Aix-en-Provence, France, Aixplorer; Super Sonic Imagine); (b) an HFUS linear-array transducer (10–22 MHz, Genova, Italy, My Lab^TM^ class C; Esaote SpA); (c) an HFUS linear-array transducer (22–38 MHz, Beijing, China, Paragon XHD^TM^ class C; KOLO); and (d) ultrasound biological microscopy (UBM) equipped with a mechanically driven linear transducer (50 MHz, Tianjing, China, MD300SII; Meda Co., Ltd.). The selection of the transducer was based on the size and thickness of the lesion in order to achieve optimal imaging. Initially, a lower frequency was used to scan the lesions for their precise location and measure each lesion’s thickest part. Subsequently, an appropriate probe frequency was selected for a detailed evaluation based on the thickness of the lesion. For extremely thin lesions, UBM was used for the scanning. All ultrasound scans were conducted per the established guidelines for diagnosing skin diseases [15].

Before the HFUS scanning, images of the lesion’s appearance were stored in a database. The sonographer assisted the patient in maintaining proper positioning to ensure the total exposure of the lesion. The transducer was placed gently and vertically on the lesion’s surface during the ultrasound scanning to observe it from the center to the margin. The parameters of depth, gain, focus, and frequency were adjusted accordingly to optimize the visualization of the lesion. Applying a copious amount of gel or a gel pad is imperative to prevent compression on the lesion caused by the transducer. The color gain, color baseline, and pulse repetition frequency (PRF) were then selected to optimize the lesion’s color Doppler flow signal visualization. Additionally, the sample frame size for the area of interest was adjusted to be slightly larger than the lesion size.

### 2.3. HFUS Features Analysis

Considering the retrospective nature of this study, measures were taken to mitigate potential bias from the subjective evaluator (Q Wang) and its subsequent impact on image interpretation. Therefore, we selected two other experienced musculoskeletal sonographers, W.-W. Ren and D.-D. Shan, who had received training in the same evaluation standard for image analysis. The assessors blindly evaluated the images without prior knowledge of the pathological findings, and the results of the two assessors were consistent.

The following ultrasound features have been defined according to previous studies [3,15]. (a) Lesion size (including length and width): When visualized clearly, the length represents the maximum diameter of the lesion when measured in the longitudinal section, and the width represents the maximum diameter measured in the transverse section. (b) Thickness: The thickness should be measured in the thickest part of the lesion. (c) Shape (regular or irregular, with the former including the creeping form): The term “irregular”, in relation to skin lesions, refers to those with needle-like edges that penetrate the surrounding dermis or subcutaneous tissue. Conversely, a “regular” lesion is characterized by smooth edges without infiltration into the adjacent tissue [16]. (d) Morphology of the surface (concave, flat, and protuberance). (e) Boundary (Figure 2) of the base (well-defined and ill-defined). (f) Boundary (Figure 2) on both sides (well-defined and ill-defined). (g) Internal echogenicity (Figure 3): (homogeneous, heterogeneous). Homogeneous: the echogenicity was similar at all points in any direction of the mass. Heterogenous: the echogenicity distribution of each point in any direction of the mass was not consistent. (h) Echogenicity (isoechogenicity, hypoechogenicity, and mixed echogenicity). (i) Stratum basal (dermis, subcutaneous). (j) Doppler vascularity pattern (absent, sparsity, and profusion) (Figure 4).

### 2.4. Statistical Analysis

The data were analyzed using the Statistical Package for the Social Sciences (version 26.0; SPSS Inc., Chicago, IL, USA). Normality was assessed using the Shapiro–Wilk test. Continuous variables are described as the mean ± SD if normally distributed and analyzed using the independent samples *t*-test. Continuous variables with skewed distributions were described as medians across interquartile ranges (IQR, 25% to 75%) and were analyzed using the Wilcoxon rank-sum test. The remaining clinical and HFUS characteristics of categorical variables were compared using the Chi-square test or Fisher’s exact probability test. After the univariate analysis, a binomial logistic regression analysis was used to confirm the significant ultrasound features of high-risk DF and low-risk DF differentiation in the cohort. Finally, discrimination models were built based on statistically significant predictors. The cutoff value for the discrimination model was 0.05. In addition, the prediction model’s sensitivity, specificity, and accuracy were calculated for the cohorts.

## 3. Results

### 3.1. Patient Characteristics

In this study, a total of 50 lesions from 49 patients were collected (Figure 5). There were 17 cases of high-risk DFs, including 6 (12.0%, 6/50) cellular DFs, 6 (12.0%, 6/50) deep DFs, 4 (8.0%, 4/50) aneurysmal DFs, and 1 (2.0%, 1/50) mixed DF (aneurysm combined with atrophy). On the other hand, there were 33 cases of low-risk DFs, including 23 (46.0%, 23/50) classic DFs, 5 (10.0%, 5/50) atrophic DFs, 3 (6.0%, 3/50) lipid DFs, 1 (2.0%, 1/50) hemosiderin DF, and 1 (2.0%, 1/50) epithelial DF.

In this study, most DF patients were female, accounting for 64.0% (32/50), while 36.0% were male (18/50). The median age was 34.5 years (IQR, 28–45 years). The lesions were located on the limbs in 60.0% of patients (30/50), on the trunk in 32.0% (16/50), and on the head and neck in 8.0% (4/50).

On the other hand, there were no significant differences in age, sex, and lesion location between the high-risk DF and low-risk DF groups (all *p* > 0.05; Table 1). This finding is consistent with the results of previous studies.

### 3.2. HFUS Characteristics

The HFUS characteristics of high-risk and low-risk DFs are shown in Table 2. The thickness of a high-risk DF was greater than that of a low-risk DF (4.1 mm, IQR, 3.2–6.1 mm vs. 3.0 mm, IQR, 1.3–4.2 mm, *p* = 0.018). Compared with the low-risk DFs, the high-risk DFs mostly showed an irregular shape (70.6% vs. 21.2%), nonhomogeneous internal echogenicity (76.5% vs. 21.2%), and subcutaneous involvement of the stratum basal layer (64.7% vs. 21.2%; all *p* < 0.05). Meanwhile, CDFI signals were present in more high-risk DF lesions than low-risk DF ones (64.7% vs. 30.3%).

However, there were no significant differences in the length, width, surface morphology, boundary of the base, boundary on both sides, and echogenicity between high-risk DFs and low-risk DFs (all *p* > 0.05; Table 2).

### 3.3. Multivariate Logistic Regression Analysis

Based on Table 3, HFUS features, including the thickness, shape, stratum basal, heterogenous echogenicity, and Doppler vascularity pattern, were used as indexes in a binary regression analysis to determine the independent predictors. After the multivariate analysis, the following three independent predictors were determined: it was shown that an irregular shape (OR = 9.023; 95% CI: 1.540–52.861; *p* = 0.015), the subcutaneous involvement of the stratum basal layer (OR = 11.077; 95% CI: 1.849–66.349; *p* = 0.008), and heterogenous internal echogenicity (OR = 5.479; 95% CI: 1.051–28.570; *p* = 0.044) were independent risk factors for DF.

The risk score (RS) for each DF was determined based on the following criteria (Table 3): RS = 2.200 × (if the lesion had an irregular shape) + 2.405 × (if the basal layer was located in subcutaneous soft tissue) + 1.701 × (if internal echogenicity was nonhomogeneous) − 5.658.

With these significant features, a logistics regression equation was established as follows:$$P=\frac{exp(RS)}{1+exp(RS)}$$

The predicted probability value, P, for each DF was calculated using the abovementioned formula. The receiver operating characteristic (ROC) curve was established to determine the optimal cutoff point for *p*, which was found to be 0.54 when distinguishing high-risk DF from low-risk DF in the prediction model. The DFs with a *p*-value greater than 0.54 were classified as high-risk DFs, resulting in a sensitivity of 84.8%, specificity of 70.6%, negative predictive value of 88.5%, positive predictive value of 58.3%, and accuracy of 74.0% (Figure 6).

## 4. Discussion

Dermatofibromas (DFs) with a typical clinical appearance are easy to diagnose, whereas atypia or variants pose diagnostic challenges [17,18,19,20]. The treatment plan for patients with DF can vary depending on their risk level. However, no previous study on the differential methods for DF risk stratification exists. Therefore, finding a convenient and effective method to distinguish high-risk DFs and low-risk DFs is urgent and essential. Based on our findings, high-risk DFs are more irregular, more likely to be involved in subcutaneous tissue, and more heterogeneous in their echogenicity on HFUS. The sensitivity, specificity, and accuracy of the above three HFUS features for identifying high-risk DFs were 84.8%, 70.6%, and 74.0%, respectively. Based on these results, the HFUS application contributes to individualized and comprehensive clinical management.

Similar to previous studies [3], DF was confined to the dermis [21]. At the same time, 36.0% of DF penetrated the dermis and extended into subcutaneous tissue, which is more common in high-risk DFs in this study [22]. The HFUS showed that a high-risk DF was deeper and more irregular than a low-risk DF; former lesions were larger from superficial to deep layers (Figure 7). The above phenomena are more meaningful for skin tumor axial extensions (including tumor layer involvement and thickness) than lateral extensions. Layer involvement significantly correlates with tumor recurrence and prognosis, while tumor thickness reflects tumor volume to some extent [23,24,25]. Exploring the characteristics of the axial extension of lesions is the advantage of HFUS. HFUS has excellent advantages in assessing the transverse and axial extension of lesions. Therefore, HFUS can clarify lesions’ size, boundary, relationship with surrounding tissues, and internal blood supply, which helps evaluate the invasiveness of the lesion [26]. HFUS compensates for the shortcomings of internal morphological features that dermoscopy cannot observe [10] and overcomes the depth limitation of other noninvasive diagnostic methods, such as RCM and OCT [27]. According to the studies conducted by Vincenzo Ricci, high-frequency ultrasound has demonstrated its utility in evaluating skin masses at various depths in clinical practice. Implementing a standardized layer-by-layer ultrasound scanning technique holds great potential for improving the diagnosis of skin masses and ensuring diagnostic repeatability [16].

The internal echogenicity of high-risk DFs is heterogeneous, which could be related to their internal mixed histological components. Histologically, aneurysmal DFs are characterized by a focal interstitial hemorrhage and hemosiderin deposition [21,28,29]. Cellular DFs show marked cellular pleomorphism with small focal central necrosis in about 10% of cases [5,23,30], the latter of which may concern more aggressive tumor types. Deep DFs are usually composed of spindle-shaped cells arranged in a hierarchical pattern and surrounded by thick collagen bundles [6]. The heterogeneity of DF’s pathological components for the abovementioned types can determine the heterogeneous pattern of its internal echogenicity on HFUS.

Regarding the blood supply to the lesion, unlike the histopathological results, only 85% of aneurysmal DFs in dermoscopy have vascular structures dominated by punctate vessels with a diffuse distribution [1]. This may be due to the limited depth of dermoscopy, which prevents the observation of vascular-like structures in the deep part of the lesion. By contrast, blood supply could be observed in all aneurysmal DFs in this study, of which 60% of lesions were rich while 40% were rare. Applying HFUS in conjunction with multiple transducers enables a comprehensive evaluation of the tumor at both superficial and deep levels. In some aneurysmal DF, the blood flow signal may not be readily apparent, potentially attributed to irregular luminal structures within blood vessels and impeded blood flow caused by the compression of endothelial cells [31].

Furthermore, this study showed a significant difference in blood flow between high-risk DFs and low-risk DFs. Overall, 64.7% (11/17) of high-risk DFs had blood flow either rich or rare, whereas only 69.7% (23/33) of low-risk DFs had blood flow that was all rare. However, this difference was not statistically significant in multivariate analysis.

There are no previous studies evaluating the diagnostic performance of HFUS in DF. In this study, we analyzed the HFUS characteristics of DFs in different pathological types and established a predictive model for the risk stratification diagnosis of DF. Based on our findings, HFUS can help improve the diagnostic ability of DFs at different risk levels and provide valuable information for treatment options. For DF patients with cosmetic needs and low-risk DFs, according to the initial HFUS assessment, regular follow-up or non-surgical treatment can be performed. The treatment strategy should be changed during the follow-up period if the lesion suddenly becomes enlarged or other adverse signs appear. However, treatment options such as surgery are recommended for patients with high-risk DFs in the initial ultrasound evaluation.

Although this study provides valuable insights, some limitations should be acknowledged. First, this study is a retrospective single-center study with a relatively small sample size. Due to the diversity of treatment methods and good biological behavior, the surgical resection rate of DF is low. Therefore, although the incidence of DF is not low, relatively few specimens of DF confirmed by pathology have been obtained. Second, all HFUS features with significant clinical implications are subjective rather than quantitative. Third, we could not determine recurrence and metastasis rates in high-risk DFs due to our cases’ lack of follow-up results. Therefore, more extensive multicenter studies are needed to identify more high-risk and low-risk DF features in analyses. Furthermore, our study was examined by a single sonographer, and then the other two sonographers performed a second reading, so there are inevitably consistency issues. We will focus on this issue in the follow-up research.

In conclusion, HFUS has a specific guiding role when differentiating between high-risk and low-risk DFs. Regarding HFUS, irregular morphology, subcutaneous tissue involvement, and heterogeneous internal echogenicity are critical information for identifying high-risk DFs.

## Figures and Tables

**Figure 1 diagnostics-13-03305-f001:**
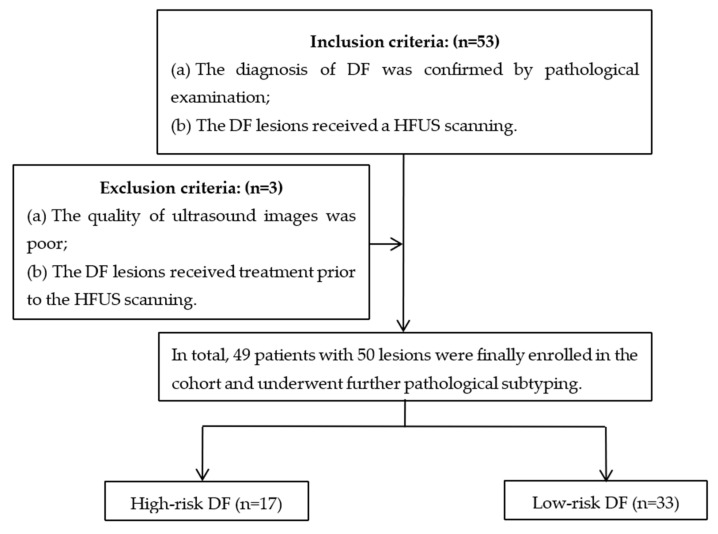
Flowchart of patient selection for dermatofibromas.

**Figure 2 diagnostics-13-03305-f002:**
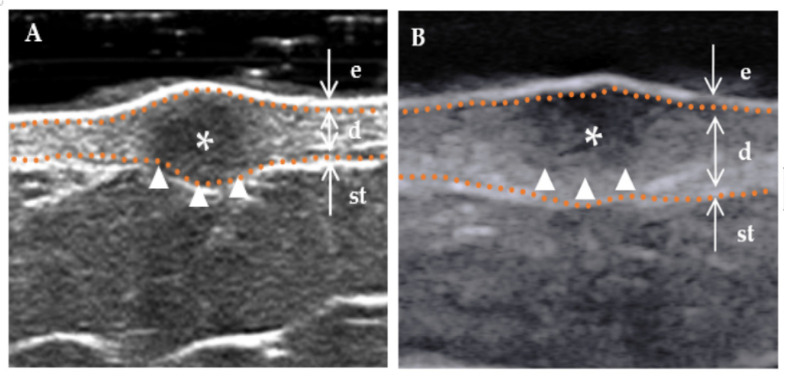
Boundary of a DF lesion. * Indicates the precise location of the lesion on the ultrasound image. The skin layers are indicated by dotted orange lines, with arrows indicating a hyperechoic epidermis (e), an isoechoic dermis (d), and a subcutaneous tissue (st). The boundary of the lesion (indicated by a white triangle) is distinctly demarcated. (**A**) “Regular” lesions are characterized by smooth edges without infiltration into the adjacent tissue. (**B**) “Irregular” lesions are characterized by jagged edges that can penetrate the surrounding dermis or subcutaneous tissue.

**Figure 3 diagnostics-13-03305-f003:**
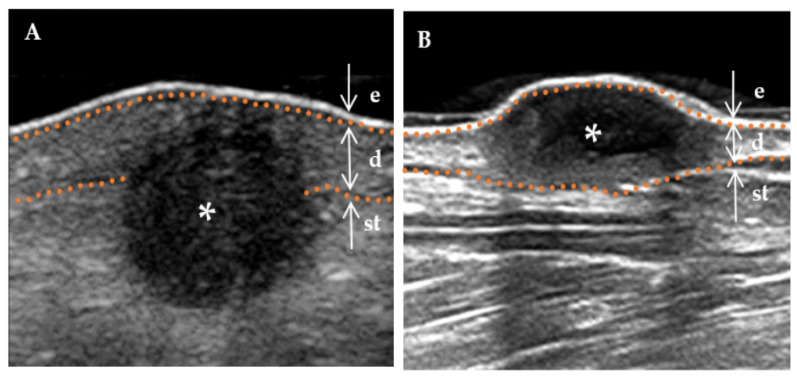
The internal echogenicity of DF lesions. * Indicates the precise location of the lesion on the ultrasound image. The skin layers are indicated by dotted orange lines, with arrows indicating a hyperechoic epidermis (e), an isoechoic dermis (d), and a subcutaneous tissue (st). (**A**) The internal echogenicity of the mass is homogeneously distributed: the mass exhibits a homogeneous low echogenicity pattern with similar echogenicity distribution in any direction. (**B**) The internal echogenicity of the mass is heterogeneously distributed: the core of the mass is hypoechoic compared to the cortex (cortical portion), which is more echogenic.

**Figure 4 diagnostics-13-03305-f004:**
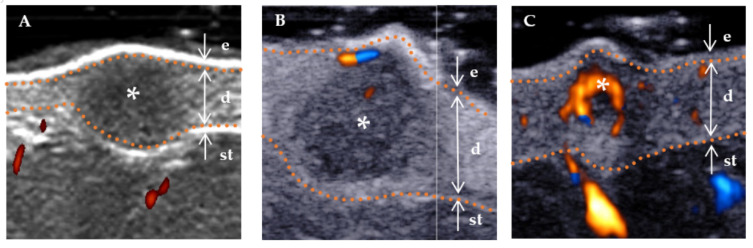
Color Doppler flow grading using high-frequency ultrasound. The color red is used to indicate the direction of blood flow towards the transducer, while blue is used to indicate the direction of blood flow away from the transducer. * Indicates the precise location of the lesion on the ultrasound image. The skin layers are indicated by dotted orange lines, with arrows indicating a hyperechoic epidermis (e), an isoechoic dermis (d), and a subcutaneous tissue (st). (**A**) No blood flow signal (absent), that is, no blood flow inside the lesion. (**B**) Rare blood flow signals (sparsity), that is, a less than two punctate and/or short rod-like blood flow, and the rod blood flow did not exceed the radius of the lesion. (**C**) Rich blood flow signals (profusion), that is, more than three punctate vessels or long vessels, could be seen penetrating the lesion, and the length could be close to or beyond the radius of the lesion.

**Figure 5 diagnostics-13-03305-f005:**
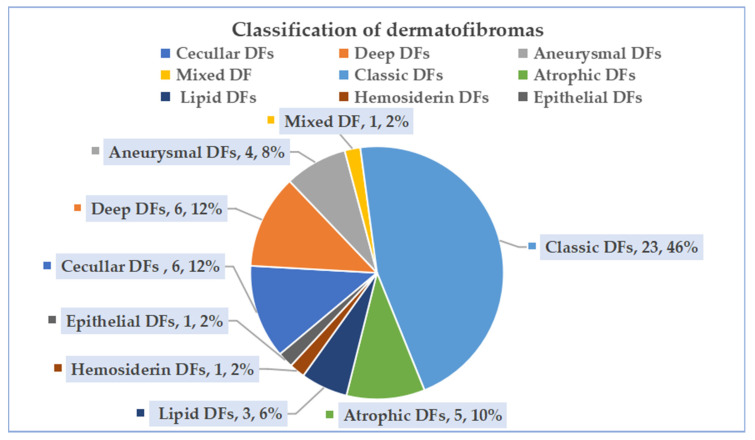
Classification of dermatofibromas.

**Figure 6 diagnostics-13-03305-f006:**
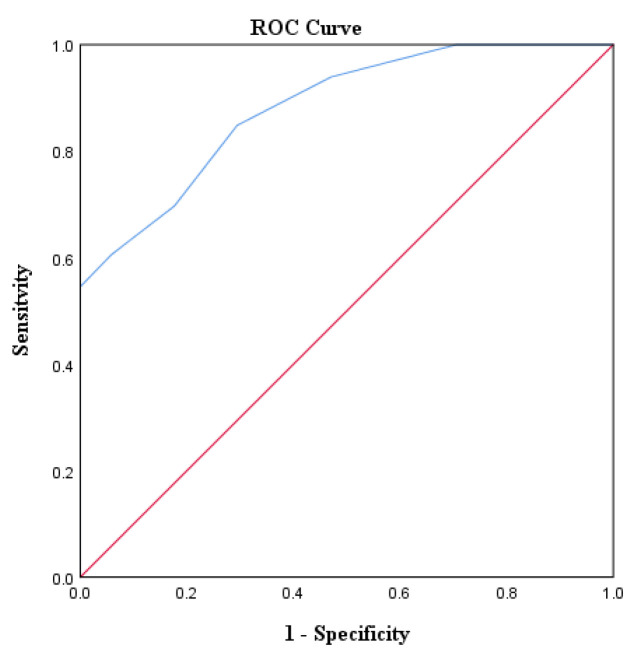
Receiver operating characteristic (ROC) curves of the prediction model (AUROC = 0.881). The red diagonal lines depict the reference lines, while the blue lines illustrate the ROC curves derived from the model.

**Figure 7 diagnostics-13-03305-f007:**
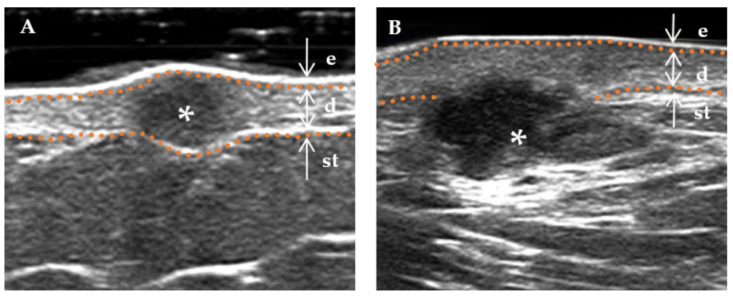
Typical high-risk DF and low-risk DF. * Indicates the precise location of the lesion on the ultrasound image. The skin layers are indicated by dotted orange lines, with arrows indicating a hyperechoic epidermis (e), an isoechoic dermis (d), and a subcutaneous tissue (st). (**A**) A 32-year-old female presenting with a hemosiderin DF measuring 5.5 × 3.2 mm, exhibiting a hypoechoic area with a clear boundary and regular shape, predominantly located within the dermis. (**B**) A 36-year-old male patient presenting with a hemosiderin DF on the face, with a mixed echo zone of 15.1 × 6.5 mm. The lesion exhibits an unclear boundary and irregular shape, extending beyond the dermis into the subcutaneous tissue layer.

**Table 1 diagnostics-13-03305-t001:** Patient demographics and clinical characteristics.

	High-Risk DF	Low-Risk DF	*p*-Value
	(*n* = 17)	(*n* = 33)	
Age ^a^ (years)	38 (25–53)	34 (31–45)	0.782
Sex, *n* (%)			0.941
Male	6 (35.3%)	12 (36.4%)	
Female	11 (64.7%)	21 (63.6%)	
Location, *n* (%)			0.725
Head and neck	2 (11.8%)	2 (6.1%)	
Trunk	6 (35.3%)	10 (30.3%)	
Limbs	9 (52.9%)	21 (63.6%)	

Note: DF, dermatofibroma. Data are the number of patients or lesions unless otherwise stated. ^a^ Expressed as the median with interquartile ranges in parentheses.

**Table 2 diagnostics-13-03305-t002:** HFUS features of high-risk DF and low-risk DF.

	High-Risk DF	Low-Risk DF	*p*-Value
	(*n* = 17)	(*n* = 33)	
Lesion size(mm)			
Length ^a^	8.8 (5.4–12.6)	6.5 (4.7–7.7)	0.061
Width ^a^	7.0 (4.6–10.9)	5.8 (4.2–7.5)	0.122
Thickness ^a^ (mm)	4.1 (3.2–6.1)	3.1 (1.3–4.2)	0.018 *
shape, *n* (%)			0.001 *
Regular (incl. creeping)	5 (29.4%)	26 (78.8%)	
Irregularity	12 (70.6%)	7 (21.2%)	
Morphology of surface, *n* (%)			0.335
Concave	0 (0.00%)	2 (6.1%)	
Flat	7 (41.2%)	16 (48.5%)	
Protuberance	10 (58.8%)	15 (45.4%)	
Boundary of the base, *n* (%)			0.773
Well-defined	7 (41.2%)	15 (45.5%)	
Ill-defined	10 (58.8%)	18 (54.6%)	
Boundary on both sides, *n* (%)			0.191
Well-defined	2 (11.8%)	11 (33.3%)	
Ill-defined	15 (88.2%)	22 (66.7%)	
Internal echogenicity, *n* (%)			0.004 *
Homogenous	4 (23.5%)	27 (81.8%)	
Heterogenous	13 (76.5%)	6 (21.2%)	
Echogenicity, *n* (%)			0.195
Isoechogenicity	1 (5.9%)	2 (6.1%)	
Hypoechogenicity	13 (76.5%)	30 (90.9%)	
Mixed echogenicity	3 (17.6%)	1 (3.0%)	
Stratum basal, *n* (%)			0.002 *
Dermis	6 (35.3%)	26 (78.8%)	
Subcutaneous	11 (64.7%)	7 (21.2%)	
Doppler vascularity pattern, *n* (%)			0.038 *
Absent	6 (35.3%)	23 (69.7%)	
Sparsity	8 (47.1%)	9 (27.3%)	
Profusion	3 (17.6%)	1 (3.0%)	

Note: DF, dermatofibroma. Data are the number of patients or lesions unless otherwise stated. * A *p*-value < 0.05 means the difference was statistically significant. ^a^ Expressed as the median with interquartile ranges in parentheses.

**Table 3 diagnostics-13-03305-t003:** Multivariate analysis in establishing the discrimination model.

Parameters	B	SE	OR Value	95% CI	*p*-Value
Shape	2.200	0.902	9.023	1.540–52.861	0.015 *
Stratum basal	2.405	0.913	11.077	1.849–66.349	0.008 *
Internal echogenicity	1.701	0.843	5.479	1.051–28.570	0.044 *
Constant	−2.925	1.019	0.054		0.004 *

Note: Numbers in parentheses are 95% CIs. Abbreviations: CI, confidence interval; SE, standard error; OR value, odds ratio. * A *p*-value < 0.05 means the difference was statistically significant.

## Data Availability

The datasets generated and analyzed during the current study are not publicly available but are available from the corresponding author upon reasonable request.

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
