# Peer review of "High-Frequency Ultrasound Imaging to Distinguish High-Risk and Low-Risk Dermatofibromas"

_diagnostics, 2023, doi:10.3390/diagnostics13213305_

Round 1

Reviewer 1 Report

The manuscript describes the role of high-frequency ultrasound imaging in assessing dermatofibromas and differentiating low-risk and high-risk patterns. It can be interesting for the pertinent scientific literature but revisions are necessary prior to publication. 

Title

I suggest revising the title as follows:

"High-frequency Ultrasound Imaging to Distinguish High-Risk and Low-Risk Dermatofibromas".

Introduction

"HFUS can clearly define lesions in superficial and deep layers of the skin".

We strongly agree with the authors but we suggest further describing the growing role of HFUS in assessing the dermo-epidermal complex and the subcutaneous tissue both in normal and pathological conditions. Indeed, nowadays the medical definition of "sono-histological pattern" is progressively mounting considering the ability of high-frequency probes to accurately distinguish the different layers and pathological changes. Standardized protocols to perform a layer-by-layer "sono-dissection" of the different superficial tissues are also adopted by plastic surgeons to accurately plan the surgical approach. Please, refer to J Plast Reconstr Aesthet Surg. 2022 Oct;75(10):3877-3903. doi: 10.1016/j.bjps.2022.08.035.    

2.2. HFUS Examinations 

"Then, the color Doppler flow signal (CDFI) should be adjusted to eliminate noise and visualize the blood flow within the lesion."

Any additional tips and tricks in order to accurately set the ultrasound machine for the color Doppler assessment? Pulse repetition frequency? How to position the region of interest to avoid artifacts? Technical pitfalls?

2.3. HFUS Features Analysis 

"Shape (regular or irregular, with the former including the creeping form)"

I suggest better describing the shape of the different lesions. For instance, in the pertinent literature, the "irregular" shape of skin lesions is described with spiculated margins infiltrating the surrounding dermis or subcutaneous fat tissue. Likewise, a "regular" shape is considered with smooth edges displacing but not infiltrating the surrounding tissues. Please, refer to Pathol Res Pract. 2022 Sep;237:154003. doi: 10.1016/j.prp.2022.154003. 

Figure 3B

The authors can clarify in the corresponding legend that the core of the mass is hypoechoic compared to the cortex (cortical portion) which is more echogenic. 

Figures (all)

For all the figures, if possible the authors should describe in the corresponding legends the exact location of the lesion (e.g., dermis, subcutis, both, etc.). In this sense, the spatial description is paramount for a better understanding and reproducibility by the readers. Some abbreviations can be added over the sonographic image - e.g., subc: subcutaneous tissue, d: dermis, etc.

Likewise, if the hyperechoic dermo-hypodermal interface is preserved or interrupted by the mass should be specified because this sonographic sign is paramount to the differentiation between high-risk and low-risk DFs.

Figure 5 should be revised to avoid the overlapping of different elements.

Minor grammatical errors should be checked and revised throughout the text.

4. Discussion 

"Based on our findings, high-risk DFs are more irregular, more likely to be involved in subcutaneous tissue, and more heterogeneous in their echogenicity on HFUS."

"36.0% of DF penetrated the dermis and extended into the subcutaneous tissue, with this phenomenon being more common in high-risk DFs in this study" 

The authors should add in the manuscript sonographic images that clearly demonstrate the subcutaneous involvement in high-risk DFs and the location within the dermo-epidermal complex of the low-risk DFs. I mean, sonographic images of dermo-hypodermal interface involvement should be added.

 Minor editing of the English language required

Reviewer 2 Report

1. The authors can avoid using unnecessary abbreviations in the abstract and throughout the MS. Generally, a use of at least three times would be appropriate to decide before the abbreviation is constructed.

2. Why were the US images assessed by different authors than the single one who had actually performed the US exam ? This is quite atypical if someone is to call the imaging as 'US examination'. The authors need to comment on this and perhaps also add it in their limitation section.

3. Why are there no clinicians or pathologists among the authors ? Their presence might have added extra to the discussion which - in its current form - lacks clinical relevance.

4. The authors can read/cite the following reference for enriching their discussion. Ricci V, et al. Pathol Res Pract 2022;237:154003. 

-

Reviewer 3 Report

The manuscript addresses the relevant topic of high-frequency ultrasound evaluation of dermatofibromas. There are few minor issues requiring authors' attention:

- Methods: please report intra-examiner reliability for the physician who performed the US scans and the inter-examiner concordance for the radiologists who assessed the images. 

- Please report negative and positive predictive values along with the sensitivity/specificity analysis.

- High-frequency ultrasound application has been widely reported in the literature. The authors are encouraged to check for previous similar studies to set comparison with their results (e.g. see 10.1111/jdv.16583)

A review by a native English speaker is suggested to improve  the quality of the manuscript

Round 2

Reviewer 1 Report

The revised manuscript can be accepted for publication in the journal.

Minor editing of English language required